# Design and Psychophysical Evaluation of a Novel Wearable Upper-Arm Tactile Display Device

**DOI:** 10.3390/s23104909

**Published:** 2023-05-19

**Authors:** Yongqing Zhu, Peter Xiaoping Liu, Jinfeng Gao

**Affiliations:** 1School of Information Science and Engineering, Zhejiang Sci-Tech University, Hangzhou 310018, China; 13262037213@163.com (Y.Z.); jfgao@zstu.edu.cn (J.G.); 2Department of Systems and Computer Engineering, Carleton University, Ottawa, ON K1S 5B6, Canada

**Keywords:** tactile display device, multi-sensory tactile, skin sensory channels, just noticeable difference

## Abstract

A novel wearable upper arm tactile display device, which can simultaneously provide three types of tactile stimuli (i.e., squeezing, stretching, and vibration) is presented. The squeezing and stretching stimulation of the skin is generated by two motors simultaneously driving the nylon belt in the opposite and the same direction, respectively. In addition, four evenly spaced vibration motors are fixed around the user’s arm by an elastic nylon band. There is also a unique structural design for assembling the control module and actuator, powered by two lithium batteries, making it portable and wearable. Psychophysical experiments are conducted to investigate the effect of interference on the perception of squeezing and stretching stimulation by this device. Results show that (1) different tactile stimuli actually interfere with the user’s perception compared to the case where only one stimulus is applied to the user; (2) the squeezing has a considerable impact on the stretch just noticeable difference (JND) values when both stimuli are exerted on the user, and when the squeezing is strong, while the impact of stretch on the squeezing JND values is negligible.

## 1. Introduction

Haptic (kinesthetic and tactile) sensation is the fifth sense of human beings and is also the most complex one. Touch receptors are distributed throughout the entire body to receive sensations such as temperature, humidity, pain, pressure, force and vibration from the outside world. Tactile sensation, in a narrow sense, refers to the skin sensation caused by the stimulus that gently touches the skin’s tactile receptors; while the broad sense of tactile sensation also includes the skin sensation caused by increased pressure to deform the skin, that is, pressure sensation, generally referred to as “tactile pressure sensation”.

With development of technology, people can obtain the feature information of objects by means of tactile simulation devices, such as sensors, through which they can feel the shape [1], roughness [2], depth [3], or slip of objects [4,5]. It is also possible to reproduce the sense of touch and interact with the surrounding environment through devices, such as tactile display devices. A tactile display device allows users to feel and reproduce the sensation of tactility through a series of actions, such as force and vibration [6]. With the support of such devices, people can enjoy a sense of realism and immersion in the game world [7,8]; ensuring the timely and accurate completion of operational tasks in the field of teleoperation [9]; providing tactile navigation for firefighters in low-vision environments in disaster relief missions; enabling silent command in the battlefield; in the field of car driving, it can provide tactile warning for drivers and passengers to improve driving safety [10]; and enabling blind people to access a large amount of information, read newspapers [11], walk and execute simple self-care through tactile functions [12], which greatly improves their quality of life and reduces the burden on society.

While most of the current research projects on kinesthetic/force and tactile display devices are still in their infancy and experiments are performed in laboratories and research centers, some devices with high stability and accuracy can provide a wide range of forces, such as Omega (Force Dimension, Nyon, Switzerland), Phantom (Sensable Technologies Inc., Woburnm MA, USA) [13,14], and the 6-DOF parallel tactile device designed by Vu et al. [15]. Nevertheless, because of their large sizes or weights, they are relatively bulky and not easy to move, which significantly reduces their applicability. A wearable tactile device that is small, portable, convenient, and comfortable, and does not interfere with the wearer’s movement has become a new research hotspot. The wearability allows tactile devices to be used in daily life, and they conform to a person’s body parts, which can transmit information to the wearer in a natural and intuitive way.

Most of current wearable tactile devices take advantage of skin sensory channels, such as lateral skin stretch, pressure or vibration, which can be sensed by mechanoreceptors in the skin. Skin receptors are capable of detecting a wealth of stimulus information, as well as being widely distributed throughout the body, making the skin the perfect channel for devices to communicate with humans. Furthermore, there is a low activation threshold for skin receptors, allowing for the design of small, lightweight, and inexpensive devices [16,17,18]. The skin transverse stretch device typically provides a slight skin shearing sensation utilizing a no-slip contact between the end-effector and the skin. Such a device can be rocker-based [19], linear [20], or rotary [21]. With the high sensitivity of human skin to tangential stretching, the skin transverse stretching device can provide directional information to the user. The pressure is usually applied through an actuator, which drives the band against the skin, resulting in the displacement of the band, and it, in turn, creates a sensation of squeezing and twisting. Driving methods of the squeezing devices differ, depending on the driving method of the squeezing belt. A single actuator is most commonly used to tighten the band from one side, but this creates a squeezing stimulus while also creating a stretching stimulus to the skin. Vibrating devices usually consist of vibrators, which are widely used because of their small and lightweight shape and the ability to drive at different frequencies and amplitudes. The vibrating device which has been used in many different scenarios is designed to provide navigation information and contact acceleration feedback. Uchiyama et al. [22] developed a vibrotactile glove which provides vibrational signals through a 3 × 3 array of vibrotactile actuators placed on the back of the hand, providing orientation and spatial representation to wheelchair users with severe visual impairment. Kurita et al. [23] employed vibrotactile stimulation to improve tactile sensitivity. Results also showed that applying white noise vibration on the fingertip side improved two-point discrimination, texture discrimination, and grip optimization. The vibratory devices are also used for remote control [24], outdoor navigation [25], enhanced cinematic experience [26], guidance for the visually impaired [27,28], and virtual surgery [29,30], among others. Although these tactile devices can be considered wearable, their force feedback is limited to vibrations, which themselves provide a limited complexity of sensations that do not realistically mimic human touch and can only serve as cues, thus limiting their potential to simulate a richer range of force patterns, and vibrational stimuli are not suitable for integration into small devices because vibrators typically require large spatial spacing to be recognized.

There is a tendency in recent years to integrate more tactile cues into the same device, which allows the device to present multiple types of skin sensations sequentially or simultaneously, such as squeezing and tap; squeezing and rotate; rotate and vibrate; and stretch, squeezing, and rotate simultaneously. Baumann et al. [31] evaluated two devices integrated in the same demonstration platform, a mechanical device contracting the wristband to simulate the sensation of a hand squeezing the wearer’s wrist, which is capable of contracting and relaxing at different speeds and holding any position; the other mechanical device connecting the S3114 servo to a small tapping finger with a foam tip which replicates the sensation of a finger tapping or tapping on the wearer’s wrist. Casini et al. [32] designed a lightweight and simple wearable device (CUFF) which utilizes pressure associated with normal force and stretch signals associated with tangential force to provide distributed mechanical tactile stimulation to the user’s arm skin. Dunkelberger et al. [33] proposed a novel, multi-sensory, wearable tactile actuator multi-sensory interface of stretch, squeezing, and integrated vibration elements (MISSIVE) that integrated three types of tactile actuators, which were vibrator strips, radial squeezing strips, and rocker-based lateral skin stretch actuators which could simultaneously generate vibration, radial squeezing, and lateral skin stretch sensations. Aggravi et al. [34] designed a wearable tactile device for the arm, which provides pressure, vibration, and skin stretching tactile feedback to the user’s forearm, driving an elastic fabric band wrapped around the arm by two servo motors, and four vibrotactile motors mounted at the waistband. Meli et al. [35] proposed a novel wearable skin stretching device for the upper extremity, called the “hBraclet”. The device consists of four servo motors and a linear actuator, which provides the pressure and stretching and longitudinal displacement cues that are associated with normal, tangential, and longitudinal forces, respectively, by controlling the tension and distance of two belts in contact with the skin to produce distributed mechanical tactile stimulation on the arm.

Multiple actuators are usually used in the above devices, and because of the wearability, the actuators are close to each other, which impairs the wearer’s ability to perceive the discrimination of each tactile signal [36]. A number of tactile modalities can be integrated into a single system in order to reduce the spacing between actuators and the perceptual interference between different types of signals, but, by doing so, the type of tactile sensations presented is reduced, as well as the variety of information presented.

In this paper, we designed a novel upper-arm-based multi-sensory wearable tactile display device. Psychophysical experiments on tactile stimulus perception were also conducted utilizing this device with the constant stimulus method. The effects on the perception of squeezing stimulation and stretching stimulation in the presence of interfering stimulation were investigated by quantifying the recognizability, stimulus salience, and perceptibility of skin squeezing and stretching when presented alone or simultaneously.

The novelty and contributions of this work exist in at least four folds in addition to its electrical and mechanical designs. (1) A novel mechanism and design for generating tactile stimulation is presented. Specifically, when two motors rotate in the opposite direction, squeezing stimulation is generated, and when two motors rotate in the same direction, stretching stimulation is provided; (2) three different tactile sensations (i.e., squeezing, stretch, and vibration) can be provided simultaneously or individually, depending on specific applications; (3) the device is light-weight, less than one pound, because of its novel structural design for assembling the control module and actuator; (4) psychophysical experiments were performed to evaluate the effects of multiply sensory stimuli on the user and these findings about the just noticeable difference (JND) and threshold values are very helpful for choosing appropriate stimulus ranges when multiple tactile stimuli are required to be applied on users in order to reduce their interference with each other.

The rest of the paper is structured as follows. In Section 2, the design and implementation of the new tactile device are described in detail. The designing of the experiments is described in Section 3. The results of the experiments are presented and discussed in Section 4. Finally, this paper is summarized in Section 5 along with some new recommendations for future work.

## 2. Tactile Device Design

The device, a multi-sensory wearable tactile display device, is integrated with three tactile modules, squeezing, stretch, and vibration, which can work independently or in combination to provide different types of stimuli at the same time. In the design, the device chose a micro-motor and lightweight 3D printing materials to reduce weight; the entire device size is 145 mm × 90 mm × 62 mm, weighing 368 g, and the bottom of the device uses an ergonomic curved structure, so that the device is better worn on the left upper arm, with good portability and operability.

The 3D model of the wearable haptic display device is shown in Figure 1. The device consists of three modules, namely the squeezing module (A), the stretch module (B), and the vibration module (not shown in the 3D figure), and is controlled by an Arduino UNO board (G) and powered by a battery box (C) containing two lithium batteries. Among them, the squeezing and stretch modules are fixed to the 3D printed housing (E), which rotates the I-beam wheel (D) by driving the servo motor (F), which, in turn, causes the nylon band to apply the desired stimulation to the skin. The bottom of the shell is curved to better fit the skin. There is ample space inside the housing for the Arduino UNO board (G) and for storing the wiring. The vibration module is secured directly to the forearm by an elastic fabric strap.

The physical object is shown in Figure 2. The device is fixed to the upper arm by a 5 cm wide elastic cloth belt (J). When the device starts working, two servo motors of the squeezing module (A) rotate in opposite directions at the same time to make the non-elastic nylon belt (I) on both sides tighten at the same time; two servo motors of the stretch module (B) rotate in the same direction at the same time to make the non-elastic nylon belt (I) release while tightening; four vibration motors of the vibration module (H) are distributed in the upper, lower, left, and right of the user’s arm, which can be started at the same time or separately.

### 2.1. Working Principle

Like the hRing wearable finger device proposed by Pacchierotti et al. [18], the design is based on the idea that when both servo motors rotate in opposite directions at the same time, the nylon belt is tightened to provide a force $F_{N}$ perpendicular to the skin (left side of the figure) as shown in Figure 3. Alternatively, when both servo motors are rotated in the same direction at the same time, the nylon band exerts a shearing force $F_{f}$ on the skin (right side of the figure).

The servo motor is position-controlled for this device, which means that it can only be made to work by controlling the angle of rotation of the motor. The relationship between the rotation angle of each motor and the displacement of the nylon belt is
(1)$$\Delta \;d_{i}=r\;\Delta \;\theta_{i}\;i=1,2,3,4$$
where *r* is the radius of the servo motor pulley, the radius of the four servo motor pulleys in the device is the same, all are 8 mm. $\Delta \;d_{i}$ representing the displacement of the nylon belt caused by the motor work. $\Delta \;\theta_{i}$ is the rotation angle of the *i*-th motor, expressed in radians.

Each two servo motors rotate by the same amount when the servo motor starts working, i.e., $\left|\Delta \;\theta_{1}\right|=\left|\Delta \;\theta_{2}\right|$, $\left|\Delta \;\theta_{3}\right|=\left|\Delta \;\theta_{4}\right|$, $\left|\Delta \;d_{1}\right|=\left|\Delta \;d_{2}\right|$ and $\left|\Delta \;d_{3}\right|=\left|\Delta \;d_{4}\right|$, then
(2)$$\left\{ \begin{array}{c} \Delta \;d_{p}=2sgn\left(\Delta \;\theta_{1}\right)\Delta \;d_{1}\;if\;sgn\left(\Delta \;\theta_{1}\right)\neq sgn(\Delta \;\theta_{2})\\ \Delta \;d_{l}=sgn\left(\Delta \;\theta_{3}\right)\Delta \;d_{3}\;if\;sgn\left(\Delta \;\theta_{3}\right)=sgn(\Delta \;\theta_{4}) \end{array}\right.$$
where $\Delta \;d_{p}$ indicates the displacement of the nylon belt produced by the servo motor rotating in the opposite direction, with positive values indicating a tightening of the nylon belt and negative values indicating a loosening of the nylon belt. $\Delta \;d_{l}$ denotes the displacement of the nylon band produced by the rotation of the servo motor in the same direction, with positive values indicating extension toward the outside of the arm and negative values toward the inside. The device provides vibrotactile feedback at four equidistant points on the arm in addition to these two types of stimuli, which allows the system to easily convey directional information, i.e., up, down, left, and right.

### 2.2. Device Design

#### 2.2.1. Squeezing Module

The squeezing module (A) consists of two servo motors (MG90S steering gear, Shenzhen Xintai Microelectronics Technology Co., Ltd., Shenzhen, China), two 3D printed I-wheels (yellow green high toughness resin, accuracy 0.1 mm, error 0.2%), and inelastic nylon belt for providing extrusion action. The steering engine (F) is a motor which can achieve a fixed angle of deflection by inputting pulse width modulation(PWM) with specified duty cycle. Different angles of deflection can be achieved by adjusting the duty cycle of the input PWM wave, and different models of servos can achieve different angles of deflection. MG90S steering gear is selected in this paper, its torque is 2 KG/cm, reaction speed is 0.11 s/60 degrees (4.8 V), weight is 13.6 g, gear is copper teeth, has the advantages of large torque force, high speed, gear will not break teeth due to excessive load, and can achieve a 0${}^{\circ }$–180${}^{\circ }$ deflection. The I-beam wheels (D) are fixed with the steering engines through the cross rocker. The ends of the nylon belt (I) with a width of 15 mm are wound around the two I-beam wheels which plays a fixed role. When the two servos are activated, the belts on both sides will be wound around the I-beam wheels at the same time according to the rotation of the servos, thus realizing the simultaneous tightening of the belts on the user’s arms on both sides to produce a displacement of the nylon belt $\Delta \;d_{p}$, which produces a radial squeezing stimulus at the bottom of the arms, and its operating effect is as shown in Figure 4.

#### 2.2.2. Stretch Module

Similar to the squeezing module, the stretch module (B) also consists of two servo motors (MG90S steering gear, Shenzhen Xintai Microelectronics Technology Co., Ltd., Shenzhen, China), two 3D printed I-beam wheels (D), and inelastic nylon belt (I) for stretching action, which is also controlled by the Arduino UNO development board (G). However, the difference with the squeezing module is that, when the two servos are activated, it drives the two I-beam wheels to rotate in the same direction. In other words, one of the nylon belts on the two I-beam wheels is released and the other is tightened and wound, which, in turn, causes lateral displacement of the belt on the user’s arm $\Delta \;d_{l}$, resulting in lateral shear stimulation of the part of the skin in contact with the nylon belt, making it called the stretch module. The running effect is shown in Figure 5.

#### 2.2.3. Vibration Module

The vibration module (H) consists of four miniature vibration motors (1034, Shenzhen Kuanyang Intelligent Electronic Technology Co., Ltd., Shenzhen, China), which are fixed to the top, bottom, left, and right sides of the user’s arm by elastic nylon straps, evenly distributed at 90 degrees. Given that the circumference of each user’s arm is different, the position of each vibration motor can be adjusted according to the circumference of the user’s arm while wearing it, to ensure that the vibration stimulation can be accurately felt by each user at the top, bottom, left, and right side of the arm when the palm is facing down.

## 3. Experimental Design

The sensory threshold of the device was tested in order to provide significant tactile stimulation to the user and to maximize the number of stimuli that the wearable tactile display device could provide the user. A constant stimulus method was used to measure the difference threshold, also known as JND, which is the minimum change in the intensity of a stimulus that one can perceive [37]. It is a psychometric measure of the difference between two stimulus intensities. The two cutaneous tactile signals, skin squeezing and stretch, were used as stimuli in this experiment, and the subjects were required to compare multiple stimulus pairs consisting of standard and comparison stimuli. An extensive pilot experiment was first conducted to determine the range of comparison stimuli used for the experiment. The maximum intensity of the selected stimulus was the intensity that could be felt by all subjects, which means that the probability that the stimulus amount could be felt by the subjects should be about 95%. The minimum intensity of the selected stimulus was the value that could barely be felt by all subjects, which means that the probability that the stimulus amount could be felt by the subjects should be no more than 5%. The difference between the stimulus amounts in the selected stimuli should be equal and the number of presentations of each stimulus should be equal with no less than 50.

According to the characteristics of the constant stimulus method, a large number of pilot tests were carried out to determine the best range of stimulus intensity. Secondly, we selected seven intensities of stimulus within the defined range of stimulus intensity as comparison stimulus, namely 37°, 47°, 57°, 67°, 77°, 87°, and 97° (with the motor rotation angle as the reported amount), while the intermediate intensity stimulus was taken as the standard stimulus, i.e., 67°. All stimuli were presented to the subjects in random order, and each comparison stimulus was compared to the standard stimulus 50 times, with 350 comparison experiments performed in one set of experiments, making a total of four sets of experiments. Stimuli presented were compared by the subjects and reported verbally.

While testing the JND, this experiment also evaluated whether there were interference effects in the transmission of the two types of tactile signals under this tactile device. The four sets of experiments were squeezing stimulus perception test alone, squeezing the stimulus perception test with stretch interference, stretch stimulus perception test alone, and stretch stimulus perception test with squeezing interference. The tactile device was placed on the subject’s left upper arm at the beginning of each set of tests (Figure 6), and the nylon straps of both modules were adjusted until good contact was maintained with the subject’s arm, with the subject being informed of the test for each set of tests. The subjects were asked to focus their attention on the main stimulus of the test, close their eyes, wear noise-canceling headphones, and play pink noise to isolate visual and auditory influences as much as possible during the experiment. A “1” is reported when the first stimulus is perceived more clearly, and is accordingly recorded as a “+”; a “2” is reported when the second stimulus is perceived more clearly, and is accordingly recorded as a “−”; “3” was reported when the two stimuli were perceived as equal, and was accordingly recorded as “=”. The stimulus presentation interval was 1 second in each group of stimuli, the time spent in each experiment was approximately 5 s, the time spent in each group of tests was about 30 min, not more than 40 min. The subjects were asked to rest for 30 min at the end of each group of tests, and then the next group of tests was conducted in order to minimize the fatigue effect.

There were 18 subjects in this experiment, 8 males and 10 females, with an average age of 27 years, all right-handed. The subjects were in good health, had normal tactile perception and cognition, with informed consent to the experiment.

## 4. Simulation and Results

The data collected were processed and analyzed after the completion of the experiment, which was to calculate the proportion of a subject’s response to one comparative stimulus in each condition by the recorded responses presented with each group of stimuli, which in turn was used to calculate the 50% JND for the subject in the condition by linear interpolation. The proportion of responses P was calculated as follows.
(3)$$P=\frac{\sum y_{i}}{n}$$
where $y_{i}=1$ is the number of experiments in which subjects perceived a comparison stimulus larger than the standard stimulus, and *n* is the number of experiments performed for a comparison stimulus in each test set, i.e., n=5.

The results are shown in Figure 7 and Figure 8 For each condition, scatter plots of the angle of difference from the standard stimulus on the x-axis and the proportion of detection on the y-axis were plotted and fitted to each subject’s individual data using the logit link function. It is clear from the fitting results that the fitted curve is S-shaped, of which the proportion of each response starts close to 0 and ends close to 1, in accordance with psychophysical studies.

Statistical analyses were also performed to determine the interaction between squeezing and stretching stimulus perception, and the results are shown in Figure 9. A paired *t*-test of the estimated JND values showed that the difference between the JND for the squeezing stimulus alone (Mean(M) = 9.74°, Standard deviation (SD) = 2.18°), and the JND for the squeezing stimulus under 67° stretch interference (M = 10.68°, SD = 3.46°) was not statistically significant, t(17)=−1.67, p=0.11>0.05. The difference between the JND for the stretching stimulus alone (M = 11.20°, SD = 3.24°) differed statistically from the JND of the stretching stimulus (M = 12.97°, SD = 3.55°) under 67° squeezing interference, t(17)=−2.96, p=0.01<0.05.

The results of the JND values for the squeezing stimulus alone, the squeezing stimulus with stretching interference, the stretching stimulus alone, and the stretching stimulus with squeezing interference show that the interference affects the perceptual ability of the subjects, for different objects of interference, the magnitude of this effect varies. The interference of the stretching stimulus with the squeezing stimulus was not significant, while the squeezing stimulus had a significant effect on the interference of the stretching stimulus. Compared with the stretched JND alone, the stretched JND under extrusion interference is significantly increased as we can see from Figure 9. It is probably due to different magnitudes of force produced by the squeezing and stretching stimuli, with the perceived force of the squeezing stimulus being somewhat greater, and the stretching stimulus being more inclined to the perception brought about by the rotation of the nylon band. The effect of this interference is more pronounced when the comparison stimulus is smaller than the standard stimulus and is significantly reduced when the comparison stimulus is larger than the standard stimulus, as shown in Figure 10, which indicates that the effect of squeezing interference on the perception of stretch is asymmetric. What is also worth noting is the fact that the squeezing interference stimulus we chose in this experiment is the same as the stretching standard stimulus, which is also very likely that the size of the squeezing interference stimulus will affect the perceptual ability of the stretching stimulus, which deserves further study.

## 5. Conclusions

In this work, we have designed a novel multi-sensory wearable tactile display device with dimensions of 145 mm × 90 mm × 62 mm and a weight of 368 g, which is highly portable and maneuverable. The device combines multiple types of tactile types, squeezing, skin stretch, and vibrate to provide mechanical tactile stimulation in the user’s upper arm. The stimulus acts at different locations on the upper arm, as well as these stimuli are different in terms of perceptual type, which minimizes perceptual interference and improves perceptual accuracy. By utilizing this wearable tactile display device, we also have designed a psychophysical experiment based on the constant stimulus method. The just noticeable difference in the subjects’ perception of the stimulus was quantified, and the effect of interference on the perception of squeezing stimulation and stretching stimulation was investigated. The experimental results showed that there is mutual interference between multi-sensory tactile stimulation compared to tactile stimulation alone, but the extent of the effect varies with the primary perceptual stimulation. When the squeezing stimulation was the primary perceptual skin stimulus, providing interference from the stretching stimulation did not have a significant effect on the just noticeable difference in the perception of the squeezing stimulation. Providing interference with the squeezing stimulation when the stretching stimulation was the predominantly perceived skin stimulation made the just noticeable difference in the subjects’ perception larger, which means that the presence of the interference stimulation made the subjects less able to perceive the change in stimulating intensity. Simultaneously, it was also found that the interference of the squeezing stimulation to the stretch stimulation existed mainly in the range where the intensity of the squeezing stimulation was smaller than the intensity of the stretching stimulation. In conclusion, compared with the stretching stimulation, the squeezing stimulation was more significant and the subjects’ perception of the squeezing stimulation was more resistant to interference. Therefore, when both the squeezing stimulation and the stretching stimulation exist, the intensity of the stretching stimulation should be increased to reduce the interference of other stimulation and to improve the perceptibility of the stretching stimulation.

The results of psychophysical experiments on the just noticeable difference help us to choose the right combination of stimulation ranges when designing multiple tactile signals, in order to reduce the interference of different signals among multiple tactile signals. When designing a multiple tactile device, a stronger stretching stimulation should be selected to reduce the interference of squeezing stimulation, enabling the wearer to perceive each tactile stimulation correctly and clearly, improving the perceptual recognizability of the device. The experimental results also provide a basis for designing similar wearable tactile display devices in the future to maximize the saliency of the stimulation and minimize the interference effect.

In the future, we will explore and quantify the recognizability of tactile devices. In the meantime, the device will be further improved by reducing the size and weight while improving comfortability and durability of the system, which will enhance its wearability. Furthermore, sensors may be added to the device, which is useful for specific scenarios for qualitative testing, such as navigation.

## Figures and Tables

**Figure 1 sensors-23-04909-f001:**
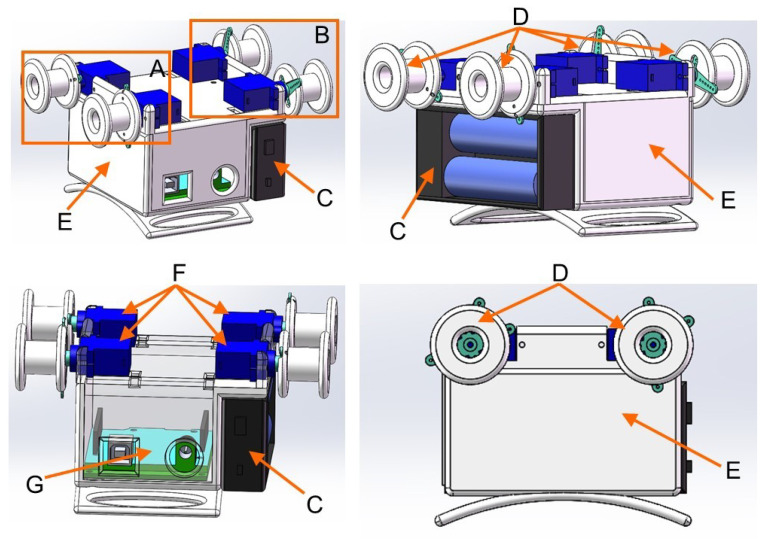
3D model of the tactile display device.

**Figure 2 sensors-23-04909-f002:**
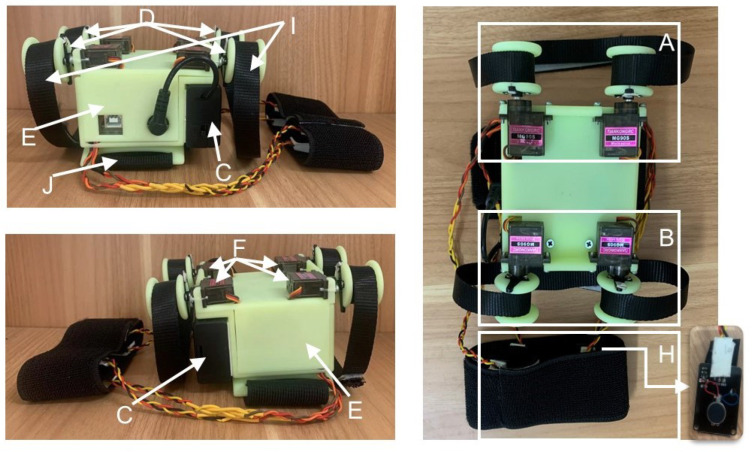
The physical object of the tactile display device.

**Figure 3 sensors-23-04909-f003:**
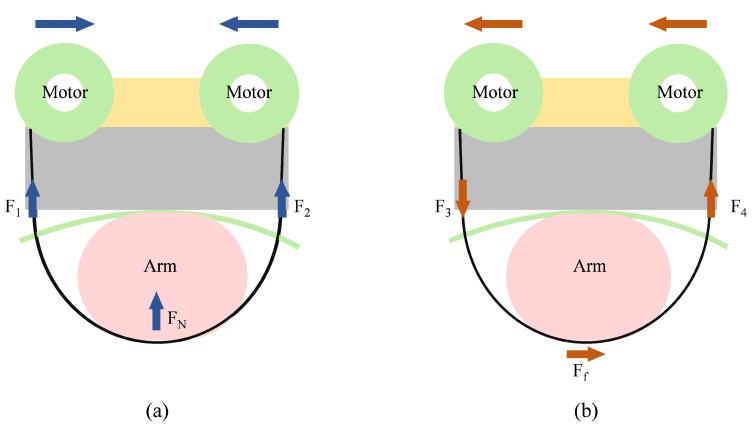
The working principle of the tactile display device. (**a**) The belt is pressed into the user’s arm as the motor rotates in the opposite direction. (**b**) When the motor is rotated in the same direction, the belt exerts a shearing force on the skin.

**Figure 4 sensors-23-04909-f004:**
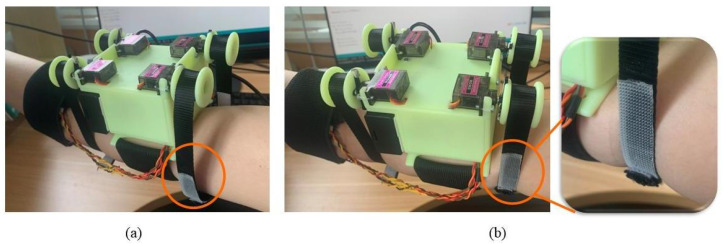
Squeezing module operation effect diagram. (**a**) When the stimulus intensity is 0°, there is no contact between the nylon band and the arm. (**b**) When the stimulus intensity is 90°, both sides of the nylon belt tighten at the same time, producing radial squeezing stimulation to the skin, and the squeezing effect is shown in the small figure on the right. (With the motor rotation angle as the reported amount).

**Figure 5 sensors-23-04909-f005:**
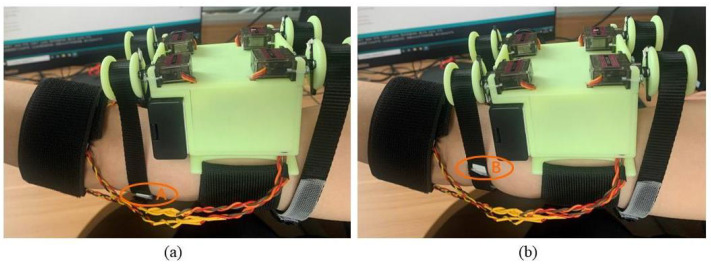
Stretching module operation effect diagram. (**a**) When the stimulus intensity is 0°, the marker point is located at A. (**b**) When the stimulus intensity is 180°, the marker point is rotated to point B. (With the motor rotation angle as the reported amount).

**Figure 6 sensors-23-04909-f006:**
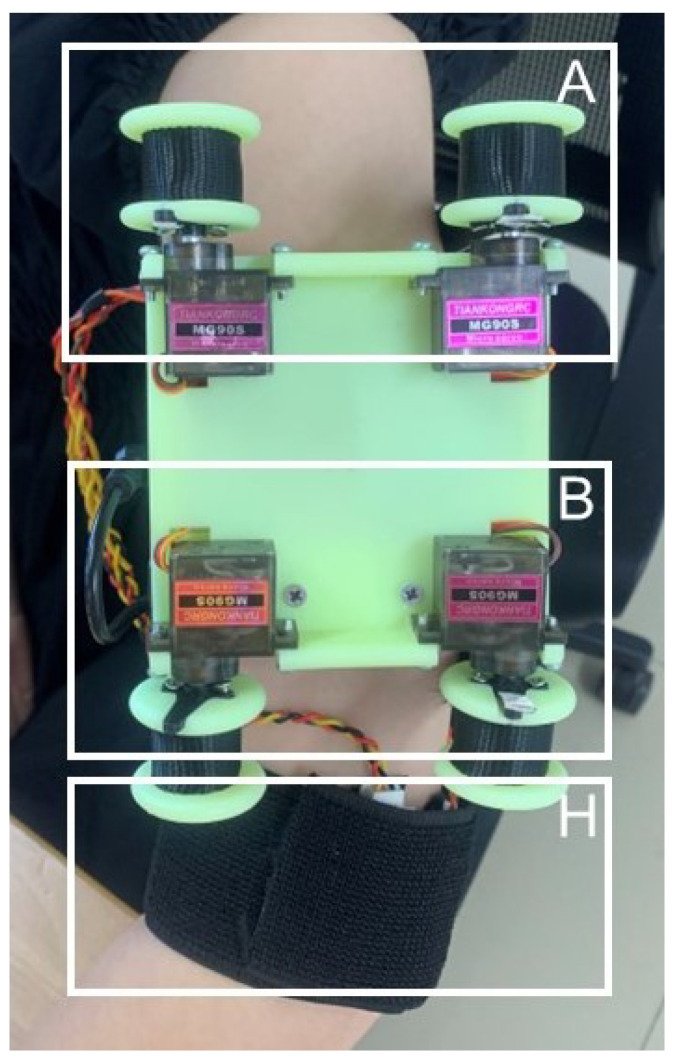
Tactile display device is worn on the upper arm of the tester. A is the squeezing module, B is the stretch module, and H is the vibration module, and the three modules can be activated simultaneously or individually.

**Figure 7 sensors-23-04909-f007:**
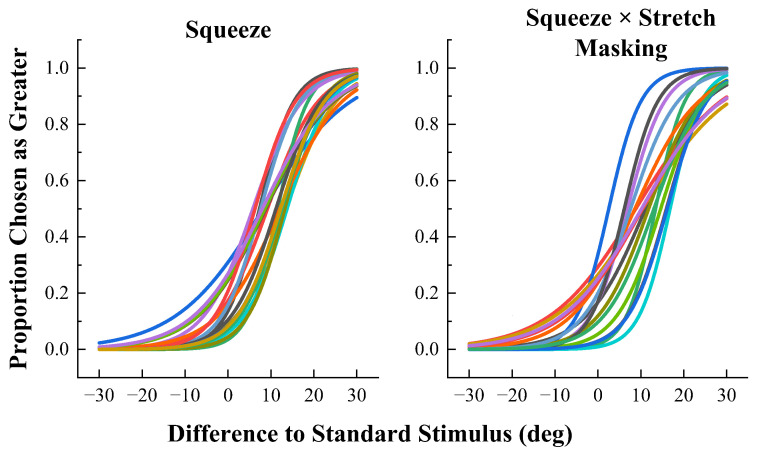
The proportional fitted curves of responses to squeezing stimulation alone and to squeezing stimulation with stretch interference conditions. The different colors represent the proportion of responses from different subjects.

**Figure 8 sensors-23-04909-f008:**
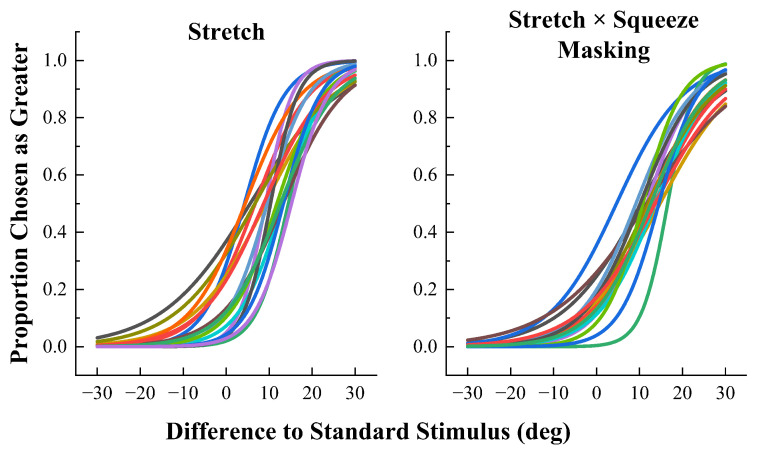
The proportional fitted curves of responses to stretching stimulation alone and to stretching stimulation with squeeze interference conditions. The different colors represent the proportion of responses from different subjects.

**Figure 9 sensors-23-04909-f009:**
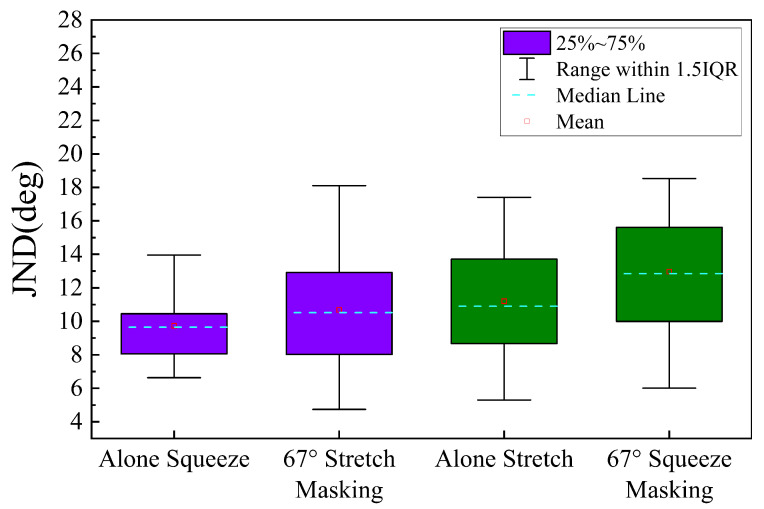
Just noticeable difference(JND) for squeezing and stretch conditions. (**Left**) No statistically significant difference between squeezing stimulus alone compared to squeezing stimulus in the stretching interference condition. (**Right**) Statistically significant difference between stretching stimulus alone compared to stretch stimulus in the squeezing interference condition.

**Figure 10 sensors-23-04909-f010:**
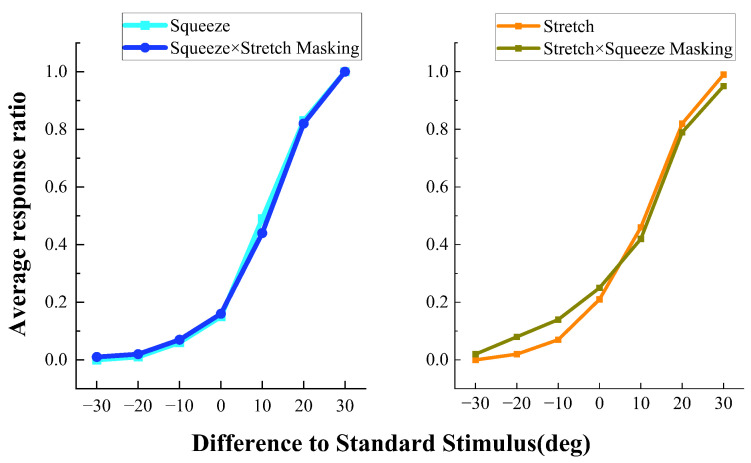
Average response ratios for squeezing stimuli, stretch interference conditions for squeezing stimuli, stretching stimuli, and squeezing interference conditions for stretching stimuli.

## Data Availability

Not applicable.

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
