# Peer review of "Design and Psychophysical Evaluation of a Novel Wearable Upper-Arm Tactile Display Device"

_sensors, 2023, doi:10.3390/s23104909_

Round 1
Reviewer 1 Report
This manuscript reports a multisensory wearable tactile display device that can provide tactile sensations of squeeze, stretch and vibration on the user’s arm. It falls within the interest of Sensors, and I have the following concerns for the authors to further consider.
1. The whole device looks bulky and may not be conformtable to wear, so can it be further improved in terms of wearability?
2. The authors should compare the advantages of current device design with previous works in a more detailed manner, for example a comparison table would help.
3. A scale bar should be provided in Figure 2 to show the actual size of the device.
4. Is there any feedback on the force applied on human arm or safety procedure to prevent the overloaded force on human arm?
5. The authors should to through the whole manuscript more carefully, since there are many typos in the manuscript.
As in Comment 5.
Author Response
Point 1: The whole device looks bulky and may not be comfortable to wear, so can it be further improved in terms of wearability?
Response 1: The size of the device is 145mm×90mm×62mm and the weight is 368g. The average length of the upper arm of an adult is 298.5mm, so the device is suitable for wearing on the upper arm. Furthermore, the bottom of the device is designed with a curved structure, which can fit the arm better.
BTW, the size and volume of the device are mainly determined by the cost and the availability of the current technology. We have tried our best to minimize the size and volume of the device based on our budget and the availability of the current technology. Anyway, improvements in wearability are the next step in our work plan, and we will consider choosing a smaller motor and a simpler structural design to further reduce the overall size of the device to improve wearability and comfort.
Point 2: The authors should compare the advantages of current device design with previous works in a more detailed manner, for example a comparison table would help.
Response 2: Thank you for your suggestion, the novelty of the device has been described in detail in the paper.
Compared to existing studies, the device uses a novel tactile stimulus generation method. In the squeeze module, when two motors are driven simultaneously in opposite directions, they drive both ends of the nylon belt to tighten simultaneously, and the bottom will generate an upward normal force on the skin, which in turn generates the squeeze stimulus perception. This stimulus generation method provides a purely normal force compared to a single actuator tightening the strap from one side. In the stretch module, when two motors are driven in the same direction simultaneously, with one end of the nylon band in a tightened state and one end in a released state, the bottom is displaced laterally around the arm, thus generating the stretch perception. Compared with the semi-circular tactile rocker, the contact area is increased, making the stretch perception more obvious, and there is the only contact with the skin without large pressure, so there is no discomfort during the actuation.
In addition, the device can present both the squeeze stimulus and the stretch stimulus simultaneously, and act on different positions of the upper arm, which can minimize the perceptual interference and improve perceptual accuracy. Furthermore, using this tactile display device, a psychophysical experiment was designed to propose the use of the constant stimulus method to measure the just noticeable difference of the subject, and to investigate the effect of interference on the perception of the squeeze stimulus and the perception of the stretch stimulus.
Point 3: A scale bar should be provided in Figure 2 to show the actual size of the device.
Response 3: Thank you for your suggestions. The dimensions of the device are 145mm x 90mm x 62mm and the weight is 368g, which has been added in the second section.
Point 4: Is there any feedback on the force applied on human arm or safety procedure to prevent the overloaded force on human arm?
Response 4: Based on experiments, it will not do harm to the subjects even the motor applies the maximum torque. So, there is no safety issue.
Point 5: The authors should to through the whole manuscript more carefully, since there are many typos in the manuscript.
Response 5: Thank you for your suggestions. We have gone through the paper a few times, trying our best to improve the writing.
Reviewer 2 Report
The work is interesting and useful for biomedical research.
However, there are some uncertainties:
- how big and heavy is the device (portable and wearable) and can it be used in current practice?
- how is the psychological component of using this device highlighted?
- it would be good, perhaps, to highlight more the difference between the combined recordings of the stretch and the squeeze or between the squeeze and the stretch.
perhaps at the conclusion more observations and discussions regarding the validity of the system and its usefulness.
Author Response
Point 1: How big and heavy is the device (portable and wearable) and can it be used in current practice?
Response 1: The dimensions of the device are 145mm×90mm×62mm and the weight is 368g. In comparison with existing similar devices, it provides good portability and operability. In addition, the size and volume of the device are mainly determined by the availability of the current technology and the cost.
Point 2: How is the psychological component of using this device highlighted?
Response 2: For the psychological aspects of using the device, we designed a psychophysical experiment using the device and measured the subject's just noticeable difference (JND) numbers using the constant stimulus method, which is the minimum amount of change in the intensity of a stimulus that one can perceive. This is a psychometric measure of the difference between two stimulus intensities.
Point 3: It would be good, perhaps, to highlight more the difference between the combined recordings of the stretch and the squeeze or between the squeeze and the stretch.
Response 3: Thank you for your suggestions and we have revised the paper accordingly. The main study in this paper is the effect of interference on the perception of squeezing stimulation and the perception of stretching stimulation, so only the comparison of the perception between stretching and squeezing in the stimulation condition alone and in the condition with interference stimulation was conducted, and the difference between the combined recordings of stretch and squeeze or squeeze and stretch was not in the scope of the original study. The comparison revealed that the squeezing stimulation was more significant and more resistant to interference than the stretching stimulation. Therefore, when both squeezing and stretching stimuli are present, the intensity of the stretching stimulus should be increased to reduce interference from other stimuli and to improve the perceptibility of the stretching stimulus. The results not only reveal the interplay and relationship between interference and perception of squeeze stimuli and perception of stretch stimuli, which provides a reference for the design of further tactile information coding, but also provide a basis for the design of future wearable tactile display devices.
Point 4: perhaps at the conclusion more observations and discussions regarding the validity of the system and its usefulness.
Response 4: Thank you for your suggestion, and we have revised the relevant parts accordingly.
Reviewer 3 Report
The aim of this paper is to develop a wearable upper arm tactile display device, which can simultaneously provide three types of tactile stimuli.
This paper is logically organized and clearly structured. However, several points should be enhanced.
Squeeze stimulation is generated when two motors rotate in opposite directions, and stretch stimulation is provided when two motors rotate in the same direction. Why? Figure 3 should be enhanced.
The tactile display device is complicated and large because many motors are used. Is it suitable for wearable? What’s the feeling of the subjects?
What’s the reliability, repeatability and wearability?
Only 9 subjects in this experiment are not enough. Please add more subjects.
Figure 7 and 8: Please indicate the different curves.
Moderate editing of English language
Author Response
Please see the attached pdf file for the response to review.

Round 2
Reviewer 1 Report
The authors have addressed the comments properly, it can be accepted now.
Author Response
Point 1: The authors have addressed the comments properly, it can be accepted now.
Response 1: Thank you for reviewing the manuscript.
Reviewer 2 Report
The paper has been improved and my comments taken into account.
I would only suggest additional proof reading.
Author Response
Point 1: The paper has been improved and my comments taken into account. I would only suggest additional proof reading.
Response 1: Thank you for your suggestion and we have gone through the paper a few times trying to clear all writing errors and typos as much as we could.
Reviewer 3 Report
The paper has been improved. However, only 9 subjects in this experiment are not enough. The authors have not addressed the comments properly.
Moderate editing of English language
Author Response
Point 1: The paper has been improved. However, only 9 subjects in this experiment are not enough. The authors have not addressed the comments properly.
Response 1: Thank you for your suggestion. In this review, we have doubled the number of subjects in the experiment and the article has been revised accordingly.
Point 2: Moderate editing of English language.
Response 1: Thank you for your suggestion and we have gone through the paper a few times trying to clear all writing errors and typos as much as we could.